# What You See is What You Classify: Black Box Attributions

**Steven Stalder**
Swiss Data Science Center
ETH Zurich, Switzerland

**Nathanaël Perraudin**
Swiss Data Science Center
ETH Zurich, Switzerland

**Radhakrishna Achanta**
Swiss Data Science Center
EPFL, Switzerland

**Fernando Perez-Cruz**
Swiss Data Science Center
ETH Zurich, Switzerland

**Michele Volpi**
Swiss Data Science Center
ETH Zurich, Switzerland

## Abstract

An important step towards explaining deep image classifiers lies in the identification of image regions that contribute to individual class scores in the model's output. However, doing this accurately is a difficult task due to the black-box nature of such networks. Most existing approaches find such attributions either using activations and gradients or by repeatedly perturbing the input. We instead address this challenge by training a second deep network, the *Explainer*, to predict attributions for a pre-trained black-box classifier, the *Explanandum*. These attributions are provided in the form of masks that only show the classifier-relevant parts of an image, masking out the rest. Our approach produces sharper and more boundary-precise masks when compared to the saliency maps generated by other methods. Moreover, unlike most existing approaches, ours is capable of directly generating very distinct class-specific masks in a single forward pass. This makes the proposed method very efficient during inference. We show that our attributions are superior to established methods both visually and quantitatively with respect to the PASCAL VOC-2007 and Microsoft COCO-2014 datasets.

## 1 Introduction

Image recognition and classification systems based on deep learning are considered black boxes. It is often hard, if not impossible, to directly locate the portions of the input image responsible for a given classification output. Being able to systematically perform such attributions is a crucial step towards explainable deep learning, which can contribute to its widespread adoption. Attribution allows understanding the potential errors and biases [13, 17, 21]. It also enables the use of deep networks as decision support tools, *e.g.* in medicine [15], and facilitating meeting legal regulations [8]. Note that we refer to attribution as the goal of associating input image regions to model decisions, which differs from the notion of interpretability that relates to models that can be directly used to infer causal or general structural relationships that link inputs and outputs [23].

The problem of performing attribution for deep image classifiers has been addressed by several recent works. These belong to two broad categories. The first relies on visualising the derived model's activations and backpropagated signals [31, 24, 3]. Although these methods provide a direct way to visualize salient areas, they heavily depend on the quality of the trained models' architectures and weights. Furthermore, some of these methods turn out to be independent of the labels the model has been trained on, as well as to some extent the parameters of the models, as shown through randomization tests [1]. This suggests that many of these methods are strongly influenced by low-level, class-agnostic features and are therefore inadequate. The second category relies on

Figure 1: **Visual comparison of *per-class* attributions** provided for VOC-2007 by our *Explainer*, alongside Grad-CAM (GCam) [24] and Extremal Perturbations (EP) [12] for the VGG-16 architecture [25]. Our attributions have sharper boundaries and at the same time are more class-accurate than Grad-CAM or EP. Attributions for only five out of the twenty VOC-2007 classes are shown for convenience. Colormap ranges from low (blue) to high (red) saliency.

local perturbations of the input images and is often model-agnostic. These models rely on masking strategies to infer local feature importance, which often results in loss of visual detail, ambiguity and high computational cost. Nonetheless, results suggest that local perturbations, like masking, are an effective way to analyze and explain the response of a model [20, 22].

This last line of reasoning posits that attribution can be formulated as an additional task complementing the classification from the *Explanandum* (*i.e.* the model to be explained). Therefore, it is natural to consider learning an *Explainer* (*i.e.* the model that explains) that specifically addresses the task of attribution. We approach attribution as a supervised meta-learning problem through the use of a complementary convolutional neural network, which learns to produce dense class-specific attribution masks, balancing preservation and deletion of information from the input image as dictated by the *Explanandum*.

The resulting *Explainer* efficiently masks and precisely locates regions of the input image that are responsible for the classification result of a pre-trained and frozen *Explanandum* (see Fig. 1 and Sup. Mat. Sec. C). We evaluate our methodology on PASCAL Visual Object Classes (VOC-2007) [9] and Microsoft Common Objects in Context (COCO-2014) [18]. To sum up, we present an efficient attribution method, which has the following advantages:

- Our *Explainer* directly provides a precise mask for each class label as in Fig. 1.
- Our *Explainer* provides attributions for test images with a single very efficient forward pass, orders of magnitude more efficient as compared to other perturbation-based approaches.
- Our *Explainer* works with any architecture of the *Explanandum*.

## 2   Related Work

**Activation and gradient-based methods.**   The methods in this category attempt to explain a trained model by leveraging its weights and its activations for a given image to be explained. Zhou et al. [31] initiated this line of research with Class Activation Mapping (CAM), where the authors built particular architectures around a notion of global average pooling and illustrated the intrinsic localization performed by a classification network. Grad-CAM [24] removed the dependency of CAM on specific architectures by reconstructing the input heatmap through a weighted average of gradients, rather than directly projecting back activations. Further extensions to this are Grad-CAM++ [3], which is CAM weighted by averaged second order gradients, Score-CAM [29], which is CAM weighted by layer-wise class activations, or Ablation-CAM [7], which is CAM weighted by neuron ablation.

Some methods directly engineer the backpropagation and network architectures in order to retrieve pixel-level saliency maps. One such example is the deconvolution approach of Zeiler and Fergus [30], where the authors propose to revert forward propagation (mapping activations to pixels instead of mapping pixels to activations) to visualize features and functions of intermediate layers. From this, Springenberg et al. [26] derived guided backpropagation. Although the focus of their work was on developing an efficient fully convolutional neural network, they show how by combining

traditional backpropagation and deconvolutions, one can retrieve salient regions of the input for a specific class. In another line of work, Wagner et al. [28] frame the problem as a defense against adversarial perturbations and tackle it by constraining and selectively pruning activations.

**Local perturbations and local models.**  The idea behind these methods is that by perturbing the input to the trained network, one can retrieve an explanation signal by summarizing changes in the classification scores. RISE introduced a way of retrieving a saliency heatmap based on random perturbations of the input image [20] and recently, [27] extended a similar reasoning for object detection systems. The input image is randomly masked several times, changes in the model scores tracked, and aggregated by a weighted average. LIME proposes a similar setup, but instead uses random switches on unsupervised segmentation [22].

Fong and Vedaldi [11] extend these ideas by reformulating the attribution problem as a meta-learning task, where the optimal mask can be found by optimizing a score. Fong et al. [12] extend their previous work [11] through the definition of *extremal perturbations*, which are the family of transformations that maximally affect a classification score. They propose to use Gaussian blur to assess perturbations over a fixed set of mask areas. Dabkowski and Gal [5] rely on a similar intuition, and the *Explainer* is directly optimized based on meta-learning around the *Explanandum*. In these settings, the *Explainer* optimizes a loss which balances preservation of the classification score on the active parts of the image, while minimizing the score for areas that are not active. These works also make use of regularization to constrain the mask area and its smoothness. In a similar fashion, Chang et al. [2] learn a pixel-specific dropout as a Bernoulli distribution, which is used to retrieve a mask and evaluate extremal perturbations. Unlike Fong et al. [12], Chang et al. [2] return the parameters of the learned per-pixel Bernoulli distribution as the saliency mask. Recently, Khorram et al. [16] proposed iGOS++, which frames attribution as optimizing over a mask which optimally balances preservation of useful information and removal of unnecessary information, regularized by bilateral total variation.

In this paper, we propose a method which learns a dense perturbation, the attribution mask, over input images. Unlike existing approaches, we optimize to balance low multi-label binary cross entropy for the unmasked portion of the input image, and high entropy for areas of the image masked out by the *Explainer*. Compared to most perturbation-based methods, our *Explainer* can predict attribution masks for all classes (and not only the predicted class) in a single forward pass, thereby being orders of magnitude faster. In addition, we beat state-of-the-art on standard benchmarks.

## 3  Training the *Explainer*

In this section, we present our *Explainer* architecture, which is trained to detect image regions that a given pre-trained classifier, the *Explanandum*, relies on to infer semantic object classes. Training of our model does *not* require any pixel-wise annotations. It relies on the *Explanandum*'s training data (including labels) and model. This makes our technique generalizable to any *Explanandum* architecture through which we can back-propagate the classification loss. In this paper though, we only consider *Explanandum* models which are trained for image classification and not for object detection or semantic segmentation.

As illustrated in Fig. 2, the *Explainer* sees an image and outputs a set of masks $\mathbf{S}$ of the same resolution, containing one segmentation mask $\mathbf{s}_c$ for each class $c \in C$. The values of the class masks $\mathbf{s}_c$ are bounded in the range $[0, 1]$ through a sigmoid function $\sigma(t) = e^t / (1 + e^t)$, where $t$ represents the predicted pixel-wise logits. Values of 1 or 0 in the mask result in the full preservation or deletion of the corresponding pixel value, respectively. Note that the sum of the class masks does not need to be equal to one.

Using the target label(s) for the input image, we split the class segmentation masks into two sets: one that consists of masks that correspond to any ground truth class appearing in the training image, while the other contains the remaining ones. We merge each set into a single mask by taking the pixel-wise maximum value for each pixel position. The target mask $\mathbf{m}$ (obtained from the first set) serves to locate the regions corresponding to (any of) the label(s) contained in a given training image, while the non-target mask $\mathbf{n}$ (obtained from the second set) allows us to gather false positive activations. Finally, the inverted mask $\tilde{\mathbf{m}} = 1 - \mathbf{m}$ is used to ensure that once the object of interest is removed, the *Explanandum* cannot address the task anymore. Note that during inference, the mask aggregation step is not needed and our *Explainer* provides one explanation mask for each class.

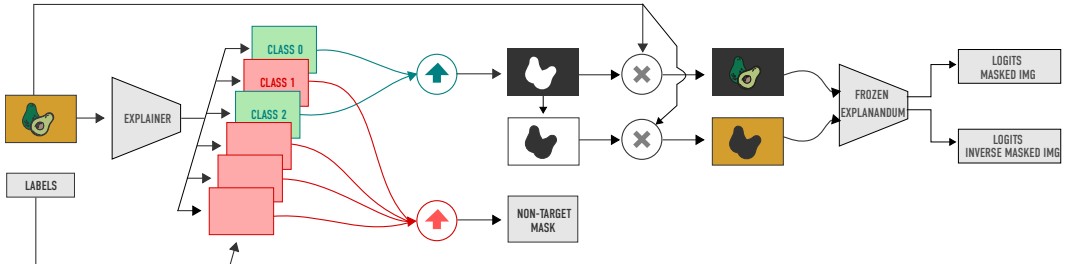

Figure 2: **Overview of our method.** Given a pre-trained *Explanandum* $\mathcal{F}$, whose weights are frozen, the *Explainer* network $\mathcal{E}$ learns to produce masks $\mathbf{s}_c$ for each class. The masks corresponding to the label(s) associated with the input image (shown in green) are merged by taking the pixel-wise maximum over masks (shown as ↑'s), to obtain a *target mask* $\mathbf{m}$ and its complement $\tilde{\mathbf{m}}$ (*i.e.* inverted mask). All the other masks (shown in red), which do not correspond to the labels of the input image but might still score positively for the given image, are merged separately to obtain the *non-target mask* $\mathbf{n}$, which is also used in the loss term. The images obtained by multiplying the target mask and its complement with the input image (shown by ×'s) are fed to the given pre-trained *Explanandum* separately, generating two outputs on which we compute losses. The set of per-class masks $\mathbf{S}$ and the aggregated *target mask* $\mathbf{m}$ serve as the attributions provided by our *Explainer*.

Note that for the selection of true classes, we chose to utilize the same ground truth labels that the *Explanandum* was given during training on unobstructed images. We believe that using ground truth label helps in handling attributions for *false negative* predictions of the *Explanandum* as we optimize the attributions also for classes which are not predicted with high probability on the unmasked images. Alternatively, one could directly take the predictions of the *Explanandum* for the selection of true classes. This requires a thresholding operation, which is an additional complication. However, relying on the *Explanandum*'s predictions might be more suited if we expect the *Explanandum* to still make a significant amount of *false positive* predictions on the training set (which was *not* the case for the architectures and datasets we decided to explain). We assume that the *Explainer* requires an *Explanandum* that performs very well on the training set, to minimize the effect of misleading training signals. Given the results in Sup. Mat. Sec. C, we are able to show that under these assumptions, this design choice does not hinder the ability of the *Explainer* to generate attributions that are faithful to the *Explanandum*'s predictions.

### 3.1 Training loss

We formulate the loss as a combination of four terms:

$$\mathcal{L}_E(\mathbf{x}, \mathcal{Y}, \mathbf{S}, \mathbf{m}, \mathbf{n}) = \mathcal{L}_c(\mathbf{x}, \mathcal{Y}, \mathbf{m}) + \lambda_e \mathcal{L}_e(\mathbf{x}, \tilde{\mathbf{m}}) + \lambda_a \mathcal{L}_a(\mathbf{m}, \mathbf{n}, \mathbf{S}) + \lambda_{tv} \mathcal{L}_{tv}(\mathbf{m}, \mathbf{n}), \quad (1)$$

where $\mathcal{L}_c(\mathbf{x}, \mathcal{Y}, \mathbf{m})$ is the binary cross-entropy loss over the masked image, $\mathcal{L}_e(\mathbf{x}, \tilde{\mathbf{m}})$ is an entropy term over the image masked with the complement of $\mathbf{m}$, $\mathcal{L}_a(\mathbf{m}, \mathbf{n}, \mathbf{S})$ is a regularization term accounting for the area of the true and false positive masks, and $\mathcal{L}_{tv}(\mathbf{m}, \mathbf{n})$ is a regularization term favouring smooth masks. $\lambda_e, \lambda_a$ and $\lambda_{tv}$ are hyperparameters balancing the loss terms.

**Classification loss: $\mathcal{L}_c(\mathbf{x}, \mathcal{Y}, \mathbf{m})$.** A model should be able to make correct decisions exclusively using the relevant portions of the image, ignoring all the rest. Under this assumption, we define the classification loss as a sum over binary cross-entropy terms, one for each class contained in the image.

Representing the *Explainer* as $\mathcal{E}$ and the frozen *Explanandum* as $\mathcal{F}$, we define $\mathbf{p} = \mathcal{F}(\mathbf{x} \odot \mathbf{m})$ as the probability vector returned by the model $\mathcal{F}$ applied on the masked image, using the aggregated (element-wise maximum) mask $\mathbf{m}$ provided by $\mathcal{E}$. For all classes $C$ in the training data and the set of target classes $\mathcal{Y}$ in the image, we compute:

$$\mathcal{L}_c(\mathbf{x}, \mathcal{Y}, \mathbf{m}) = -\frac{1}{C} \sum_{c=1}^{C} [\![c \in \mathcal{Y}]\!] \log\left(\mathbf{p}[c]\right) + [\![c \notin \mathcal{Y}]\!] \log\left(1 - \mathbf{p}[c]\right), \quad (2)$$

where $[\![c \in \mathcal{Y}]\!]$ denotes the Iverson bracket, returning 1 if the condition is true (the image contains class $c$), 0 otherwise. This allows training a mask when multiple classes are present in the training

image, as in multi-label classification problems, where multiple masks are active at the same time and some pixels are free not to belong to any mask (contrarily to using cross-entropy). Therefore, the model is free to learn dependencies and co-occurrences between different classes. This loss pushes the *Explainer* to learn masks approximating $\mathcal{F}(\mathbf{x} \odot \mathbf{m}) \approx \mathcal{F}(\mathbf{x})$, *i.e.* masked images are classified as correctly as possible in the eyes of the (pre-)trained *Explanandum*.

**Negative entropy loss:** $\mathcal{L}_e(\mathbf{x}, \tilde{\mathbf{m}})$. This part of the loss pushes the *Explainer* to provide masks whose complements do not contain any discriminative visual cue, *i.e.* parts of the image that the *Explanandum* could use to infer the correct class. In other words, the classifier scores on the mask complement should provide class probabilities $\tilde{\mathbf{p}} = \mathcal{F}(\mathbf{x} \odot \tilde{\mathbf{m}})$ as uniform as possible. To this end, we aim at minimizing the negative entropy of the predictions over the class memberships, as:

$$\mathcal{L}_e(\mathbf{x}, \tilde{\mathbf{m}}) = \frac{1}{C} \sum_{c=1}^{C} \tilde{\mathbf{p}}[c] \log \tilde{\mathbf{p}}[c] \tag{3}$$

Note that this loss is the negative entropy, as we aim for a high entropy for the background.

**Area loss:** $\mathcal{L}_a(\mathbf{m}, \mathbf{n}, \mathbf{S})$. With the terms $\mathcal{L}_c$ and $\mathcal{L}_e$ alone, our *Explainer* would have no incentive to produce a target mask that conceals image areas. Clearly, a target mask $\mathbf{m}$ with 1 everywhere would minimize these terms with respect to $\mathbf{m}$. To make sure $\mathbf{m}$ conceals background, we add two terms to the loss that correspond to two crucial desiderata: The mask should be as small as possible, but if the objective would favour larger areas, a whole range of areas in between a minimal and maximal percentage should not be penalized further.

The regularization over the area is simply formulated as the mean of the mask values, computed as $\mathcal{A}(\mathbf{m}) = \frac{1}{Z} \sum_{i,j} \mathbf{m}[i,j]$, where $Z$ is the number of pixels. We want this area to be small, but non-zero. The same is required of the non-target mask $\mathbf{n}$ as well.

The term bounding the mask area from both above and below is formulated starting from the constraint presented by Fong et al. [12]. However, instead of constraining the final image mask to a fixed area, we extend their technique to determine a valid minimum area $a$ and a maximum area $b$ for those class segmentation masks that correspond to at least one object in the image.

Given such a class segmentation mask $\mathbf{s}_c$, we flatten it to a one-dimensional vector, and then sort its values in descending order. Let $v(\mathbf{s}_c)$ be the vectorization operation and $r(v(\mathbf{s}_c))$ be the sorting of the vectorized class mask $\mathbf{s}_c$. We then define two vectors $\mathbf{q}_{\min}$ and $\mathbf{q}_{\max}$, each starting with a predefined number of 1's and padded with 0's to match the length of $v(\mathbf{s}_c)$, where the amount of 1 corresponds to the minimum $0 < a < b$ or maximum $a < b < 1$ area, respectively. We also define $\mathbf{q}_{\text{mask}}$, which corresponds to $\mathbf{s}_c$, vectorized, and sorted. Concretely:

$$\mathbf{q}_{\text{mask}} = r\left(v\left(\mathbf{s}_c\right)\right) \in [0,1]^Z, \quad \mathbf{q}_{\min} = [\mathbf{1}^{\lfloor Za \rfloor} \ldots \mathbf{0}^{\lfloor Z(1-a) \rfloor}], \quad \mathbf{q}_{\max} = [\mathbf{1}^{\lfloor Zb \rfloor} \ldots \mathbf{0}^{\lfloor Z(1-b) \rfloor}]. \tag{4}$$

To constrain the minimum and maximum area covered by the class segmentation mask, we then compute an area bounding measure $\mathcal{B}$ given by:

$$\mathcal{B}(\mathbf{s}) = \frac{1}{Z} \sum_{i}^{Z} \max(\mathbf{q}_{\min}[i] - \mathbf{q}_{\text{mask}}[i], 0) + \frac{1}{Z} \sum_{i}^{Z} \max(\mathbf{q}_{\text{mask}}[i] - \mathbf{q}_{\max}[i], 0). \tag{5}$$

where the first term penalizes the mask only if the largest $\lfloor Za \rfloor$ values are smaller than 1, while the second term penalizes the smallest $Z - \lfloor Zb \rfloor$ values greater than 0. Our goal is for the mask to cover *at least* a certain area independently of its maximum size.

The final area regularization is therefore:

$$\mathcal{L}_a(\mathbf{m}, \mathbf{n}, \mathbf{S}) = \mathcal{A}(\mathbf{m}) + \mathcal{A}(\mathbf{n}) + \frac{\sum_{c=1}^{C} [\![c \in \mathcal{Y}]\!] \mathcal{B}(\mathbf{s}_c)}{\sum_{c=1}^{C} [\![c \in \mathcal{Y}]\!]} \tag{6}$$

where $\mathcal{Y}$ is again the subset of the classes that are present in the given input image according to the given training labels. Note that the last term of (6) used in conjunction with the smoothness loss presented below favors masks with sharp transitions between 0 and 1.

**Smoothness Loss:** $\mathcal{L}_{tv}(\mathbf{m}, \mathbf{n})$. As a final requirement, we are interested in generating masks that appear smooth and free of artifacts. For this purpose, we utilize a total variation (TV) loss on the masks as used by Fong and Vedaldi [11] and Dabkowski and Gal [5], albeit we use the anisotropic variant given by:

$$\mathcal{TV}(\mathbf{x}) = \frac{1}{Z} \sum_{i,j} |\mathbf{x}[i,j] - \mathbf{x}[i+1,j]| + |\mathbf{x}[i,j] - \mathbf{x}[i,j+1]| . \tag{7}$$

where $\mathbf{x}$ is a two-dimensional array of values. The smoothness term of the loss relates to the target mask $\mathbf{m}$ and non-target mask $\mathbf{n}$ as:

$$\mathcal{L}_{tv}(\mathbf{m}, \mathbf{n}) = \mathcal{TV}(\mathbf{m}) + \mathcal{TV}(\mathbf{n}) \tag{8}$$

This smoothness term encourages our attributions to appear visually coherent.

## 4  Experiments

We present three results to establish the advantages of our *Explainer* over existing approaches. Although continuous in nature, our masks show a sharp transition from background to foreground, making it effectively binary. Additionally, our masks adhere tightly to region boundaries. Since common objective metrics may fail to convey this, we provide a brief visual comparison. The next result shows how our masks are class-specific, which provides more interpretive power than combined masks. The third result shows that our combined masks outperform existing methods on standard metrics for semantic segmentation.

Since we expect our *Explainer* to learn class-specific semantics and generate class-specific masks of the same resolution as the input image, we employ DeepLabV3 [4] with a ResNet-50 backbone, an architecture commonly used for semantic segmentation tasks. We train this model from scratch to avoid any bias, *i.e.* an information leakage from the semantic segmentation dataset from which DeepLabV3 has been pre-trained on.

For the *Explanandum*, we use VGG-16 [25] and ResNet-50 [14] models, both pretrained on ImageNet [6]. We fine-tune them on the multi-label classification tasks for VOC-2007 and COCO-2014 (i.e. on natural *unmasked* images), and then freeze the models. To this end, we use the full training set of VOC-2007 and 90% of the COCO-2014 training set for fine-tuning. To assess generalization, we use the test set of VOC-2007 and the validation set of COCO-2014, respectively. For choosing hyperparameters, we use the VOC-2007 validation set and the remaining 10% of the COCO-2014 training set. In this work, we implemented models and experiments using PyTorch [19] and PyTorch Lightning [10][1].

### 4.1  Visual Comparison

**Visually, our *Explainer* leads to sharp and meaningful masks.** Fig. 3 provides a visual comparison of how our *Explainer* attributions perform compared to the saliency maps obtained from Grad-CAM (GCam) [24], RISE [20], Extremal Perturbations (EP) [12], iGOS++ (iGOS) [16], and Real-Time Image Saliency (RTIS) [5]. For the RTIS baseline, we could not retrain their model on VOC-2007 and COCO-2014, as it was not possible to employ their architecture on multi-class classifiers. Instead, we use their loss to train our architecture. Accordingly, we fine-tuned some hyperparameters of our RTIS implementation, while leaving the remaining ones at the default values as given in their paper. For all our experiments, we resized input images to 224x224 pixels and normalized them on the mean and standard deviation of ImageNet.

Our approach provides sharp masks, which contrasts with classical saliency methods such as Grad-CAM or RISE. EP and iGOS++ provide masks which are better localized, although not as sharp as the ones produced by RTIS or our *Explainer*. Our implementation of RTIS provides sharp masks, but the overall attribution tends to be leaking over object borders. Our area constraints, in combination with the entropy term over the mask complement $\tilde{\mathbf{m}}$, enforce tight masks over object classes of different sizes. For instance, Fig. 3(2), Fig. 3(5), or Fig. 3(8) show how our masks align much more clearly with the objects. This not only holds for single-class images, but also when we have multiple

---

[1]Code is available at: `https://github.com/stevenstalder/NN-Explainer`

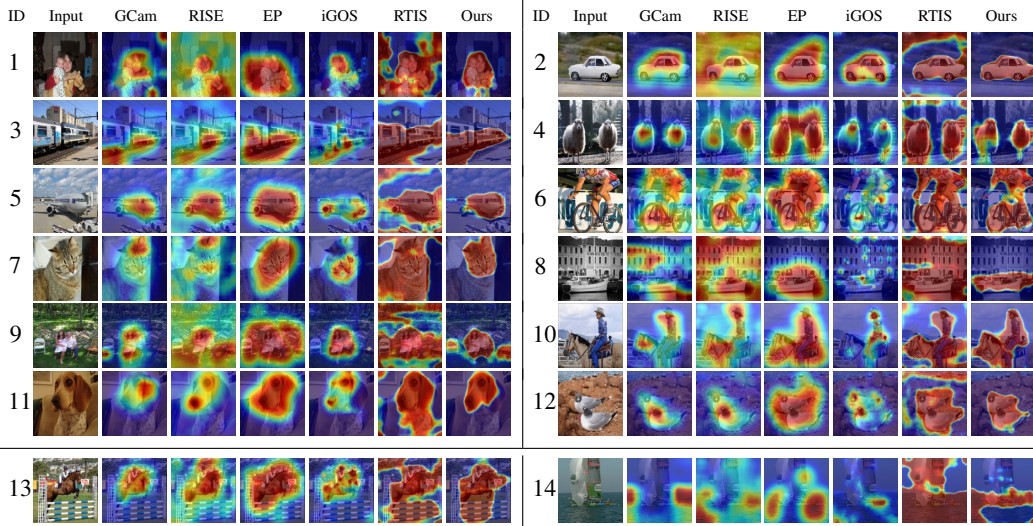

Figure 3: Visual comparison of our attributions for a VGG-16 network fine-tuned on images from the VOC-2007 dataset and frozen. Our attributions are much more effective at retaining object class regions and discarding the rest. Examples 13 and 14 show cases where our *Explainer* is inaccurate.

instances of the same class (Fig. 3(1), Fig. 3(4), Fig. 3(12)), or different classes (Fig. 3(6), Fig. 3(9), Fig. 3(10)). Attribution of some animals (*e.g.* Fig. 3(7), Fig. 3(10), Fig. 3(11)) often tends to be focusing on heads. Most training images containing animals have unobstructed heads, which can explain such behaviour. For some classes, all methods agree on the importance of an image region that is not actually part of the actual object. A good example for that is Fig. 3(3), where the train tracks are part of the attribution. Since most training images of trains also show train tracks, it seems obvious that the *Explanandum* learns to see the train tracks as discriminative semantic co-occurrence. With the help of an *Explainer*, such unintended biases can quickly be detected.

Fig. 3(13) and Fig. 3(14) show two examples where our *Explainer* is less precise. For Fig. 3(13), our attribution leaks over to the bars below the horse. Even though the classifier might indeed have made a connection between these bars and the "horse" class, Grad-CAM, RISE, EP, and iGOS++ appear to agree that the main importance should be attributed to the actual objects in the image. In Fig. 3(14), it is difficult to interpret our attribution for the class "boat" as some parts of the attribution lie on the actual object but also a significant area of the surrounding water is activated (underlining potential discriminative co-occurrence). However, the other methods also provide varying explanations, with iGOS++ masking mostly the boat, but Grad-CAM attributing all importance to the water. For more results on COCO-2014 and VOC-2007 please refer to Sup. Mat. Sec. D, E, and F.

## 4.2 Class-specific masking

**Our *Explainer* provides class-specific masks.** In Fig. 1 we have already hinted at how our *Explainer* is able to provide more class-specific masks than the other techniques. In this section we evaluate this numerically. To this end, we have selected five target classes from the VOC-2007 dataset, for which we evaluate how the class masks interact with the classification confidence of the VGG-16 *Explanandum*. We have selected the classes "cat" and "dog" as they are two animals that often occur in similar settings, the most prevalent class "person" that can often be seen together with a variety of other classes and often inside the fourth class "car", as well as the "bottle" class, to include a VOC-2007 class that is very difficult to detect for the *Explanandum*. For each test image containing one of the target classes, we first evaluate the classification performance of the *Explanandum* on the unmasked image ("none" column) and then on masked images where the image is masked once with each of the class-specific masks. Specifically, for each of these continuous masks we report the mean softmax probabilities for the target class, after we threshold each mask over nine uniform values in $[0.1, 0.9]$ at $0.1$ intervals. Note that here we use the softmax instead of the sigmoid function despite our multi-class setting since we want to evaluate the activation confidence for a single class compared

| _Explainer_ | | Mask | | | | |
|---|---|---|---|---|---|---|
| | | none | bottle | car | cat | dog | person |
| **Target Class** | bottle | 20.57 | 15.29 | 4.91 | 4.82 | 4.82 | 2.72 |
| | car | 70.52 | 6.71 | 62.92 | 6.67 | 6.66 | 7.11 |
| | cat | 72.78 | 6.09 | 6.18 | 60.55 | 6.33 | 5.74 |
| | dog | 58.00 | 4.67 | 4.70 | 5.93 | 57.60 | 4.58 |
| | person | 69.37 | 8.90 | 6.18 | 5.83 | 6.57 | 71.01 |

| _Grad-CAM_ | | Mask | | | | |
|---|---|---|---|---|---|---|
| | | none | bottle | car | cat | dog | person |
| **Target Class** | bottle | 20.57 | 16.10 | 5.24 | 4.85 | 4.76 | 3.61 |
| | car | 70.52 | 6.93 | 50.19 | 6.65 | 6.53 | 8.92 |
| | cat | 72.78 | 6.17 | 6.39 | 43.74 | 8.99 | 5.85 |
| | dog | 58.00 | 4.46 | 4.39 | 8.23 | 41.88 | 4.38 |
| | person | 69.37 | 11.81 | 8.18 | 5.90 | 8.75 | 56.38 |

| _RISE_ | | Mask | | | | |
|---|---|---|---|---|---|---|
| | | none | bottle | car | cat | dog | person |
| **Target Class** | bottle | 20.57 | 18.67 | 10.94 | 12.05 | 10.70 | 9.73 |
| | car | 70.52 | 29.75 | 56.66 | 30.27 | 34.43 | 32.42 |
| | cat | 72.78 | 37.89 | 39.28 | 49.61 | 41.89 | 36.02 |
| | dog | 58.00 | 19.64 | 20.34 | 23.72 | 45.12 | 22.83 |
| | person | 69.37 | 56.06 | 53.39 | 47.33 | 54.81 | 63.52 |

| _RTIS_ | | Mask | | | | |
|---|---|---|---|---|---|---|
| | | none | bottle | car | cat | dog | person |
| **Target Class** | bottle | 20.57 | 18.92 | 11.19 | 13.37 | 13.44 | 17.24 |
| | car | 70.52 | 41.25 | 67.02 | 40.69 | 31.87 | 54.31 |
| | cat | 72.78 | 45.84 | 36.27 | 72.62 | 59.21 | 54.16 |
| | dog | 58.00 | 41.37 | 34.76 | 57.59 | 61.22 | 58.76 |
| | person | 69.37 | 64.71 | 54.41 | 43.42 | 61.55 | 73.59 |

Table 1: Comparison of class-specific masks generated by our _Explainer_, Grad-CAM [24], RISE [20], and adapted RTIS [5]. Images containing the target class specified in the row are masked with the thresholded class masks specified in the column and given to VGG-16 for classification. In any given cell we report the mean softmax probabilities over all mask thresholds and for every test image containing the target class. If the mask class equals the target class, we want high scores, if it is different, we want low scores. We additionally compute the softmax scores achieved for unmasked images ("none" column) for reference.

to all other (ideally masked-out) classes. This can of course affect the scores for the completely unmasked images where multiple object classes may be present.

In Tab. 1, we show the evaluations for our _Explainer_, Grad-CAM, RISE and RTIS. For EP and iGOS++, it was not feasible to conduct these experiments since these methods take a very long time to generate masks compared to the other ones (see Sup. Mat. Sec. A). We can clearly see how the masks produced by our method allow the _Explanandum_ to retain very good classification confidence for the target classes when the correct mask is taken but render it completely uncertain when any other mask is put on the image. This is in stark contrast to RISE and RTIS especially, which produce overall larger, less precise masks. Even though the _Explanandum_ has high confidence after masking the images with their corresponding correct masks, the same is still true when any wrong mask is used. Only Grad-CAM achieves comparable performance to our _Explainer_ but results in considerably worse scores for the correct masks for 4 out of 5 classes and still shows some increased confidence when the wrong mask is put on the images of an often co-occurring target class (_e.g._ "cat" and "dog", or "person" and "car").

## 4.3 Segmentation accuracy

**Our attribution quality approaches object segmentation.** While our _Explainer_ architecture provides excellent attributions visually, we also confirm this quantitatively. Our explanations approximate object-level segmentation and can thus be evaluated using standard instance and semantic segmentation benchmarks. In Tab. 2 we illustrate the accuracy of the masking strategies, on VOC-2007 and COCO-2014, with VGG-16 and ResNet-50 as _Explanandum_, respectively. To provide a fair comparison with our method, we evaluate all methods by aggregating the masks from multiple classes into a single one, as shown in Fig. 3. We also provide four base measures that serve as sanity check baselines in Tab. 2), _i.e._ the columns "0", "0.5", "1", and "G.T." (ground truth). These measures show how the assessment metrics range under border conditions: with a completely null mask (composed of only 0's), with a mask composed fully of 0.5's, with a non-mask composed of only 1's and finally with a mask which is perfectly equal to the ground truth segmentation.

We employ four different metrics, namely Accuracy (Acc), Mean IoU (IoU), Saliency (Sal) and Mean Absolute Error (MAE). All metrics are computed on a per-image basis, by using the ground truth segmentation masks and the aggregated mask of the true classes present in each image, then averaged over the test set. The Accuracy is computed pixel-wise over the continuous masks without thresholding. The same holds for the IoU, where we take the pixel-wise minimum of the mask and the ground truth as the intersection and the maximum as the union, respectively. The Saliency score

| Data | Expl. | Metric | 0 | 0.5 | 1 | G.T. | GCam [24] | RISE [20] | EP [12] | iGOS++ [16] | RTIS [5] | Ours |
|---|---|---|---|---|---|---|---|---|---|---|---|---|
| VOC-2007 | VGG | ↑ Acc | 66.29 | 50.00 | 33.71 | 100 | 70.53 | 56.98 | 63.91 | 70.17 | 59.63 | 74.60 |
| | | ↑ IoU | 0 | 23.42 | 33.71 | 100 | 28.78 | | 34.95 | 24.05 | 39.94 | 38.04 |
| | | ↓ Sal | -0.85 | -0.53 | 0.15 | -0.98 | -1.41 | -0.82 | -0.77 | -1.88 | -0.31 | -1.32 |
| | | ↓ MAE | 33.71 | 50.00 | 66.29 | 0 | 29.47 | 43.02 | 36.09 | 29.83 | 40.37 | 25.40 |
| | ResNet | ↑ Acc | 66.29 | 50.00 | 33.71 | 100 | 68.54 | 57.37 | 61.43 | 69.23 | 50.77 | 73.56 |
| | | ↑ IoU | 0 | 23.42 | 33.71 | 100 | 30.75 | 28.42 | 34.88 | 24.25 | 36.46 | 34.23 |
| | | ↓ Sal | -0.67 | -0.68 | -0.08 | -1.13 | -1.29 | -1.02 | -0.77 | -1.92 | -0.35 | -1.33 |
| | | ↓ MAE | 33.71 | 50.00 | 66.29 | 0 | 31.46 | 42.63 | 38.57 | 30.77 | 49.23 | 26.44 |
| COCO-2014 | VGG | ↑ Acc | 70.39 | 50.00 | 29.61 | 100 | 66.31 | 52.35 | 57.01 | 68.70 | 39.18 | 69.90 |
| | | ↑ IoU | 0 | 20.87 | 29.61 | 100 | 25.63 | 25.50 | 29.23 | 20.35 | 31.16 | 31.59 |
| | | ↓ Sal | -0.1 | 0.03 | 0.72 | -0.38 | -0.67 | -0.31 | -0.13 | -1.12 | 0.65 | -0.51 |
| | | ↓ MAE | 29.61 | 49.36 | 70.39 | 0 | 33.20 | 47.15 | 42.55 | 30.89 | 60.76 | 29.83 |
| | ResNet | ↑ Acc | 70.39 | 50.00 | 29.61 | 100 | 67.25 | 55.05 | 56.92 | 69.57 | 41.40 | 65.30 |
| | | ↑ IoU | 0 | 20.87 | 29.61 | 100 | 24.84 | 25.98 | 29.75 | 21.63 | 30.76 | 27.42 |
| | | ↓ Sal | 0.2 | -0.25 | 0.37 | -0.88 | -0.66 | -0.44 | -0.24 | -1.19 | 0.17 | -0.64 |
| | | ↓ MAE | 29.61 | 49.36 | 70.39 | 0 | 32.36 | 44.44 | 42.68 | 30.02 | 58.46 | 34.35 |

Table 2: Segmentation scores for the attribution models. We have compared the methods against the pixel-wise segmentation ground truths of all 210 VOC-2007 test images that include such annotations as well as for a randomly selected subset of 1000 COCO-2014 images out of our entire test set of over 41K images (due to long compute times of certain methods, see Sup. Mat. Sec. A). Columns "0", "0.5", "1" and "G.T." (ground truth) represent the sanity check baselines to allow for a relative understanding of the scores. The upward arrows ↑ (downward arrows ↓) indicate that higher (lower) values are better.

is adapted from Dabkowski and Gal [5], and computed as:

$$\mathrm{Sal}(\mathbf{m}, p) = \log\left(\max(\mathcal{A}(\mathbf{m}), 0.05)\right) - \log\left(\sum_{c \in \mathcal{Y}} \mathbf{p}[c]\right), \tag{9}$$

where $\mathcal{Y}$ is the set of classes present in the image. This metric balances the size of the mask $\mathbf{m}$ with the *Explanandum* outputs. Smaller masks minimize the first term but have a tendency to lower the classification confidence, thus increasing the second term. It is therefore possible to get a lower score compared to the ground truth mask as a very small mask reduces the first term greatly, while it might still preserve a lot of the classification confidence. This is to the disadvantage of our method, as we aim to also attribute regions of lower (but still significant) importance instead of only the most distinguishing features. Finally, the MAE corresponds to the mean absolute error over the full dataset, which is computed for each input image as: $\mathrm{MAE}(\mathbf{m}, \mathbf{g}) = \frac{1}{Z}\sum_{i,j}|\mathbf{m}[i,j] - \mathbf{g}[i,j]|$, where $\mathbf{g}$ is the aggregated ground truth segmentation map for all object classes in the input image. Note that certain metrics might favor very small masks (*e.g.* Sal), while others favor very large masks (*e.g.* IoU).

The assessment in this sub-section assumes that an ideal method should perform attribution over the whole ground truth segmentation, which is not the case in certain situations, as attributions should only cover classifier-relevant parts of the object class (*e.g.* face instead of whole body). We argue that although attributions do not directly correspond to semantic segmentations, and while different methods target different goals for attribution, the combined evaluation of these metrics can still be used for a standardized assessment.

## 4.4 Limitations

Not all pairs of *Explainer* and *Explanandum* interact equally well and allow straightforward training. We observed that, while the explanation of VGG-16 and ResNet-50 on the VOC-2007 dataset performs consistently, the explanation for COCO-2014 is harder on ResNet-50 than on VGG-16 (see Sup. Mat. Sec. E and F), even though the accuracies are comparable (see Sup. Mat. Sec. B, Tab. 4). Sometimes, masks fail to precisely cover the discriminative objects parts. Some examples are given in Fig. 3(13) and Fig. 3(14), and further examples are given in the Sup. Mat. Sec. D, E and F.

# 5   Conclusions

We have presented an approach for visually attributing the classification result of a frozen, pre-trained, black-box classifier (the *Explanandum*) on specific regions of an input image. This involves training an *Explainer* architecture with the training data that was used to train the *Explanandum*. The resulting model generates attributions in the form of class-wise attribution masks, as well as an aggregated foreground mask with a single forward pass for any test image. As compared to the gradient and activation-based approaches, which often generate blurry saliency maps, our masks locate object regions more accurately and the delineation of these regions is sharp (Sec. 4.1). As compared to the approaches that use perturbed inputs, the *Explainer* is not only more precise, as we prove quantitatively (Sec. 4.2 and 4.3), but also computationally more efficient (Sup. Mat. Sec. A).

We observe that the *Explainer* can be learned well if the pre-trained classifier exhibits a good level of accuracy. Since learning the attribution fully depends on the signals provided by the *Explanandum*, this comes as no surprise. If instead of the original image, we use inputs that are masked by our *Explainer*, the accuracy may slightly decrease (Sup. Mat. Sec. B). Since our masks are so parsimonious in retaining the essential parts of the original image, it points to the fact that the drop in accuracy might come from the *Explanandum* being deprived of background-dependent biases. Therefore, once the *Explainer* is trained to generate masks, one could imagine retraining the *Explanandum* with the original as well as *Explainer*-masked data. This can ensure that classification accuracy can be both high and free of influences from non-object regions. Finally, while our work focuses on explaining deep image classifiers, we believe it should be possible to extend such an approach to other black-box classifiers, other data modalities, and other learning tasks such as regression.

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
