# OpenReview forum: "What You See is What You Classify: Black Box Attributions"
_NeurIPS.cc/2022/Conference — NeurIPS 2022 Accept_

### Official Review · Reviewer_sJ2V · 2022-07-09

**Rating:** 7
**Confidence:** 4
**Soundness:** 4 excellent
**Presentation:** 2 fair
**Contribution:** 4 excellent

**Summary:**

The paper proposes a method to identify regions of the images that identify and contribute decisively at identifying image by the decisive  object. The task is approached by using two network entitled Explanandum (i.e. the model to be explained) and Explainer (the model that explains) and training process based on 4 losses. The method is evaluated in conjunction with Pascal VOC 2007 an MS-COCO 2014 datasets

**Questions:**

1. To me the details of the problem to be solved are not firmly established. The paper says  that it uses the  "multi-label classification tasks" version of the two databases. Yet, only single class examples are shown in the evaluation. Than it speaks about "attribution" and at last about segmentation. I am missing some phrases (in different parts of the text) that:
 - say simply what the paper tries to solve
 - how can one evaluate the quality of this solution and why. The evaluation section carries the reader between different metrics and (unfortunately) excuses why some tests couldn't have been run in full.
- how can one use the heat masks found


2. The method is not explained well:
- l.104 "given pre-trained classifier"
- l 108 "Explainer sees an image and outputs a set of masks S".
 Then how the Explainer relates to the pre-trained classifier!? It seems to me that the two are independent. The first time \mathcal{F} appears  is at line 133, again it is not explained well

**Limitations:**

The paper (approached problem and solution) does not raise any ethical or societal impact. Therefore, there is no (need for) discussion from this point of view.

The paper discusses technical limitations.

**Strengths And Weaknesses:**

Strengths:
 - the idea of using another network to infer the internal mechanism of a base one is very appealing

Weaknesses:
 - on the technical side, I view this paper contribution more closely to "incremental" than "outstanding". The idea to use two networks in a differentiating analysis is known.  None of the loss terms used in optimization are new; the approach is related to image saliency and object segmentation
- given the limited contribution on the idea side, the evaluation should have been stronger. PASCAL - VOC 2007 is the smallest version, MS COCO is larger, but only parts of it were used. On the visual comparison (fig 3- main paper), the proposed method is the sharpest, but EP, with another threshold on the heat map gives very similar results.

Overall, the paper is not so strong on either side to be viewed as a clear accept.

Other small issues:
 -  If accepted, the evaluation section should be re-written to enhance the strengths (positive part). Also, the explanation of experiments details (database presentation, .. l196-205)   needs to be separated  from the actual experiments
 - figure 2 - is rather poor: Who is \mathcal{F} in the figure? \mathbf{m}? \mathbf{S}?

---

> ### Author Response · Authors · 2022-08-02
> **Response to reviewer sJ2V**
>
> ### Weaknesses
>
> _About incremental contribution and dual black box strategy._
>
> Apart from the “Real Time Image Saliency for Black Box Classifiers” paper by Dabkowski et al., over which we show significant advantages, we are not aware of any other methods using a neural network to explain another black-box network. The assessment that none of the loss terms used are entirely new is true. However, we have extended several previous ideas for the terms and have efficiently combined them in a way that allows us to retrieve more precise attributions than earlier works.
>
> _About stronger evaluation and EP attributions._
>
> It is true that EP can be equally sharp but EP (where the mask is specifically optimized for each **test** image) is also many magnitudes slower. Methods like EP were also the reason to not use the full MS COCO dataset for some experiments, as it already took more than 1 day to attribute 1000 images with EP (see Table 3 in the Supplementary Material).
>
> _Other comments:_
>
> We thank the reviewer for the additional suggestions. Regarding Figure 2, it was an oversight, the different symbols (F, m, S, E) will be added to the Figure to make it more clear.
>
> ### Questions
>
> _About multi-label training task._
>
> We disagree with the reviewer, as we show several multi-label examples – see Figures 1 and 3 (Images 6, 9, 10, and 13) as well as many examples in the Supplementary Materials – and have performed numerical evaluations on the full datasets containing both single- and multi-label examples (Table 2). Note that we refer to multi-label as the process of training a model on multiple labels per image (as opposed to one label per image), so that the Explainer can directly attribute multiple object classes for a single image at inference time.
>
>
> _About attribution and segmentation connections._
>  * _About the goal of the paper._
> The paper tries to solve the problem of generating attributions for a black-box classifier. The experiments on segmentations are only done as a means of evaluating against other methods, which we deem a reasonable assessment, despite the shortcomings of this evaluation. We discuss these limitations in lines 298-303.
>
>  * _About assessment metrics and computational time._
> Evaluating the quality of attribution methods has always been a very difficult task, which is why several other papers have proposed a wide range of evaluation metrics. We have tried to include several of these metrics to allow the reader to have a broader view on the overall quality of the different methods as there is no single standardized assessment for this task.
> Regarding the tests which could not have been run in full, we reiterate that this has been due to the slow execution speed of other methods, as it can be seen in Table 3, Section A of the Supplementary Material. While inference using VGG-16 on 1000 COCO-2014 test images takes 34s for our method, EP needs close to 34h and iGOS++ needs 26h40. Other datasets and model combinations show similar orders of magnitude of computational time. Instead of completely omitting these other baselines, we have decided that it is better to include them and to evaluate everything on a reduced dataset, to ensure a fair comparison.
>
>  * _About the use of heat maps._
> The heat maps are the final output by attribution methods and are intended to serve as visual explanations for human users.
>
> _About the dependence of Explainer and Explanandum._
>
> The classifier’s predictions are used as a backpropagated learning signal for the Explainer, making the Explainer learn from the classifier directly.

---

> ### Comment · Reviewer_sJ2V · 2022-08-08
> **Response to rebuttal**
>
> I would like to thank authors for their responses. Overall, it clarified many things. Now:
>  - my questions about evaluation have been clarified. Now I see the evaluation as being strong
>  - the paper still lacks clarity and if, accepted, my suggestion is to make an effort to improve presentation
>  - the key part: technical contribution and idea. My view is that the contribution is on a thin edge: it draws heavily inspiration from image saliency, but it presents from a perspective that, as authors emphasized, is novel. Previous saliency methods could have been used for the same application, but they have not, at least to my current knowledge. Therefore, the paper can be seen, indeed, as opening new directions.

---

### Official Review · Reviewer_wP4a · 2022-07-10

**Rating:** 5
**Confidence:** 3
**Soundness:** 3 good
**Presentation:** 2 fair
**Contribution:** 3 good

**Summary:**

The authors propose to train a deep network - the explainer to estimate attributions for a pre-trained black-box classifier. The attributions are defined as image masks, in particular two masks - the class-relevant mask and the rest. Local and smoothness constraints assist the explainer in generating crisp, highlight localized attribution masks. The authors evaluate the approach on multiple datasets and compare with standard explanation methods.


**Questions:**

Lines 29 and 30 - The authors portray the dependence of the explainability methods on trained model architectures and weights as undesirable. I however, would argue that this is desirable to enhance the faithfulness of the explanations tied to the trained model. The authors have to either elaborate more on their claim or provide ‌references supporting their claim. The subsequent sentence (line 31) is more appropriate. There is evidence suggesting that the explanations generated through certain methods appear to be independent of model parameters (Adebayo et al., and Sixt et al).

The third advantage (line 53) is confusing. If explanandum is not required, then how does one ensure the faithfulness of the explanations to the explanandum? Perhaps the authors refer to a model-agnostic explainer?

Line 65- Perhaps the citation is incorrect. Adebayo et al [1] do not handle issues arising from high sensitivity to noise and clipping effects of gradients.

Line 114 - target label(s) for the input image - are these the predicted labels of the explanandum or the ground truth labels? If it is ground truth labels, then the explainer can never be faithful to the explanandum - what is the explainer then explaining? The authors should clarify if their method is post-hoc or explainable by design?

I reiterate the initial concern on faithfulness based on the observations in lines 475-482 (supplementary material). The explainer does not seem to be faithful to the explanandum!

Furthermore, I notice almost no highlights for regions other than the top predicted class (Figures 4, 5 supplementary material). How does the model explain non-zero probabilities assigned to the other classes?


**Limitations:**

not applicable.

**Strengths And Weaknesses:**

Strengths

The ability to automatically generate reasonably accurate segmentation masks (albeit in an opaque manner) is a key strength of the approach.

The authors have experimented with multiple datasets and architectures to validate the method.

The paper is well-written and structured. It is easy to follow.

Weakness

It is interesting to note that Grad-CAM results in comparable segmentation masks without the need to train an additional mask generator (Table 1). Grad-CAM extrapolates the low dimensional mask to the original image size that adds some noisy artifacts. Unlike the proposed explainer, it does not enforce any locality/smooth constraints. Despite these artifacts, Grad-CAM a very simple approach in contrast to the proposed explainer, can generate good segmentation masks. Thus, the proposed method’s advantage is unclear.

While the authors provide a quantitative comparison of existing approaches through the segmentation dataset, as the aim is to provide a human interpretable explanation, a user study to validate the same will strengthen the contribution.

Finally, the approach reads as a blackbox (mask generator) trying to explain another blackbox (classifier).

---

> ### Author Response · Authors · 2022-08-02
> **Response to reviewer wP4a (Weaknesses)**
>
> _About the comparison with Grad-CAM._
>
> We agree that Grad-CAM is simpler. However, our approach is more accurate. It shows clearer boundaries (Figure 3) and better numbers (Tables 1 and 2). We acknowledge that in certain applications, a user might prefer Grad-CAM over our approach for its simplicity, but that in other applications having an accurate attribution is far more important than having a simpler method (e.g. in the medical domain). Furthermore, even though the Explainer requires a training process, at evaluation time it is equally fast as Grad-CAM.
>
> _About human validation through user study._
>
> It is a fair observation. We did not consider it because we are not aware of any competing method doing this.
>
> _About one black box explaining another._
>
> Yes, we have shown that we can unravel a black box using another. This is a different approach and a novelty of our contribution. We do not consider it as a weakness.

---

> ### Author Response · Authors · 2022-08-02
> **Response to reviewer wP4a (Questions)**
>
> _About explaining models using their trained weights._
>
> If we understand the reviewer correctly, the reviewer is suggesting that it is desirable to be dependent on the parameters of the model. We agree on this point. However, we argue that not accessing weights directly but by learning a model interpreting the outputs – which are dependent on the weights – is a more robust way to provide an explanation of a given pretrained classifier (the Explanandum).
>
> _About faithfulness._
>
> We design our method to provide an explanation for the output of a classifier rather than trying to explain the classifier internals itself (weights, layer order, etc). We observed that all our trained explainers are faithful in practice, as we show in Section C (containing Figures 4 and 5) in the Supplementary Material. There, we show that our explainer is actually very faithful to the classifier, as the average mask activation correlates really strongly with the probabilities by the classifier, even in cases where the classifier makes wrong predictions (e.g. rows 2,3, and 5 in Figure 4 or row 7 in Figure 5).
>
> Bellow we provide detailed responses for a few questions raised by the reviewer.
>
>  * _The third advantage (line 53) is confusing_
> Our third point needs reformulation: At inference time, our Explainer directly generates attributions for any class without needing an explicit signal from the Explanandum. However, we are aware that in practice one may want to retrieve the classification probabilities from the Explanandum as well (which is of course possible and would just need two forward passes).
>
>  * _Ground truth labels are used at training time._
> We use the ground truth labels and we acknowledge the concern about faithfulness from the reviewer as we have not sufficiently explained our incentives for doing it this way. The choice was between taking the same ground truth labels that the classifier was given during training on unobstructed images or taking the predicted labels and we chose the first option. We believe that the first approach is better at handling attributions for **false negative** predictions of a classifier since we still optimize the attributions for classes which aren’t predicted with high probability on the unmasked images. In turn, the second approach might be more suited if we expect the classifier to still make a significant amount of **false positive** predictions on the training set (which was **not** the case for the architectures/datasets we decided to explain). However, we argue that even the potential negative effect of **false positive** predictions is to some extent mitigated in our approach since we use the classification prediction of the classifier, which is not corrected with the ground truth labels. Finally, note that using the classifier predictions would require a thresholding operation, which would be an additional complication. In the end, we are aware that the Explainer requires an Explanandum which performs very well on the training set to minimize the effect of misleading training signals.
>
>  * _Lines 475-482 (supplementary material) suggesting unfaithfulness._
> These lines are indeed confusing and should be reformulated. We strongly believe that our explainer is faithful to the classifier under the assumptions that we made above. It is true however, that we cannot rule out 100% of misleading attributions but those cases should be very rare.
>
>  * _The reviewer noticed almost no highlights for regions other than the top predicted class._
> We disagree with the assessment that only the top predicted classes are highlighted. Even for probabilities of less than 10% given classifiers, the Explainer shows visible (and sometimes even very clearly localized) attributions (see rows 3, 6, 7, and 8 in Figure 4 or row 1 in Figure 5). For objects that are predicted with even lower probabilities, the learned attributions will not show significant enough values to be clearly visualized by the heatmap. Of course one could rescale the color map to also pick up these less pronounced signals from the mask, however we have chosen to use the standard mapping that other methods like Grad-CAM have used.
>
>
> _About the citation Adebayo et al [1] in line 65._
>
> This is indeed the wrong citation and will be corrected in the potential camera-ready version.

---

> > ### Comment · Reviewer_wP4a · 2022-08-09
> > **Thank you for the clarifications**
> >
> > I thank the authors for the detailed response.
> >
> > The authors have clarified the problem setting context. I would argue that this is not a post-hoc explainability model in the classical sense. Using ground truth class labels for training, the masking network adds elements that are not faithful to the base classifier. Thus, I am unsure of the practical significance of the explanation. I would urge the authors to discuss this issue in the final draft if accepted.
> >
> > Thank you for the description of the regions highlighted for labels other than the top predicted class.
> >
> > I rather view the strength of this method as an unsupervised method for object localization. It does not require any information about object bounding boxes, yet can reasonably locate the objects.

---

### Official Review · Reviewer_Aofh · 2022-07-12

**Rating:** 8
**Confidence:** 4
**Soundness:** 4 excellent
**Presentation:** 3 good
**Contribution:** 4 excellent

**Summary:**

This manuscript focuses on to provide an elegant approach to visually attributing the classification result of a frozen, pre-trained,
black-box classifier called as Explanandum on the input image(s). The author provides a solid method for the second deep network (called "Explainer") to provide a robust mask and attributions for test images. Together with solid experiments and solid baselines, the author achieves competitive results for attributions on the well-known datasets of PASCAL VOC and MS-COCO.



**Questions:**

There is a small concern I faced in this manuscript about training the explainer, but little-to-less information has been provided on the black-box classifier (Explanandum), as we could see the explainer is trained to generate the masks, the retaining of the Explanandum with original + explainer-masked data information is missing in this section. Could I please request the authors to give some views on this as I tried to identify this root cause but found no such information on this?

**Limitations:**

My assumption is that the author neglected on the part about tiny objects/having occlusions etc. I believe this methodology works great in general but general scenarios could also have tiny object and objects with occlusions, can the author please provides some views on this. Even though this approach sounds significant and I'm more inclining towards accepting this paper, as I found out this could potentially be a significant contribution to the computer vision community.

**Strengths And Weaknesses:**

Clarity:
++ The paper reads very well and provides a very good description of related work and background, motivating the problem. Even outside of the contribution of this paper, I would recommend this paper to people getting started with deep learning/understanding black-box classifiers as it provides a thorough description of the part of the pipelines it deals with.

Novelty:
++ The proposed approach formulation is concise, convincing, and novel. A seemingly reasonable approach has been conducted in this manuscript. Compared to the prior work on performing attributions for deep learning image classifiers, the current strategy involves supervised meta-learning to produce dense class-specific attribution masks, which balances the preservation and deletion of information from the input image(s)
++ The idea of incorporating the Explainer to generate masks and retraining the Explanandum with the original as well as Explainer-masked data can ensure that classification accuracy can be high and free of influences from non-object regions. A pretty decent methodology has been applied to this attribution problem by the authors on this.

Experiments:
++ There are a number of experiments performed across datasets that are extensive, fair, and provide solid foundations. The fact that the proposed methodology achieves competitive results makes me confident in the result as the implementation and experiments are sufficient and presented clearly. Additionally, the improvements are fairly consistent. Besides, in-depth analyses have been provided on the approach for different tasks and enough information has been provided via a thorough analysis of where the benefits of the approach have been obtained.

Reproducibility:
++ Thanks to the author's efforts in giving enough clarity about the given methodology, this can be replicated with some efforts. I believe this could be a significant contribution this time in NeurIPS.

---

> ### Author Response · Authors · 2022-08-02
> **Answer to reviewer Aofh**
>
> ### Questions
>
> _About the training process of the Explanandum._
>
> This might be a misunderstanding. We have not retrained the classifier with Explainer-masked data as well. This is only an idea for future work (see lines 326-328 in the paper). For this paper, the classifiers have only been (pre-)trained on the unmodified (i.e. unmasked) images from the VOC-2007 and COCO-2014 datasets.
>
> ### Limitations
>
> _About tiny objects and occlusions._
>
> The reviewer is correct that there might be a limitation for boundary-precise attribution of tiny objects as we push the mask to have a certain size between given limits. However, we see that in practice, the model is doing a very good job attributing tiny objects that co-occur with larger ones. A great example is given in Figure 7 in the Supplementary Material. In Image 8, the Explainer is able to precisely attribute a tennis ball, which only makes up a very tiny fraction of the entire image.
>
> Regarding the Explainer’s behavior with partially occluded objects, there aren’t any good examples for that in the main part of the paper since our visualizations are done with aggregated masks. However, a clear example can be found in the first row of Figure 5 in the Supplementary Material, where the attribution for the ‘car’ class only highlights car parts and does not include the people that are occluding parts of the car.  In a revised version, we agree that more details on this interesting aspect should be included.

---

### Author Response · Authors · 2022-08-02
**Acknowledgements**

We thank all reviewers for providing us with their useful feedbacks for our contribution. We will update the paper accordingly. Please find our detailed answers below.

---

### Meta-Review · Area_Chair_rjtX · 2022-08-21

**Recommendation:** Accept
**Confidence:** Certain

**Metareview:**

The paper proposes an attribution prediction approach to enhance the interpretability of DNN models. For this purpose, a second “explainer” model is used which can generality class-specific masks for the classification of relevant regions.

The reviewers have overall commended the novelty of the approach, clear writing, and detailed experiments. However, there were concerns about the training required for explainability which makes the approach relatively more computationally demanding. Given the performance is better than GradCAM, the approach still offers an advantage and a suitable alternative. The rebuttal provided further clarity and corrections in light of the initial reviews, these changes must be incorporated in the final version. The AC will also support incorporating a user study since the final goal of these attributions is to provide visual explanations for human users.

Based on the reviews and rebuttal, AC recommends accepting the paper and would like to congratulate the authors!


**Award:**

Yes

---

### Decision · Program_Chairs · 2022-09-14

Accept